# Are We All the Same When Faced with an Ill Relative? A Person-Oriented Approach to Caring Activities and Mental Health in Emerging Adult Students

**DOI:** 10.3390/ijerph19138104

**Published:** 2022-07-01

**Authors:** Basilie Chevrier, Aurélie Untas, Géraldine Dorard

**Affiliations:** 1Aix-Marseille Univ., PSYCLE, F-13628 Aix-en-Provence, France; 2Université Paris Cité, Laboratoire de Psychopathologie et Processus de Santé, F-92100 Boulogne, France; aurelie.untas@u-paris.fr (A.U.); geraldine.dorard@u-paris.fr (G.D.)

**Keywords:** young adult carers, emerging adulthood, higher education, psychological distress, subjective well-being

## Abstract

Dealing with the illness/disability of a relative is part of everyday life for many students, who may have to take on caring responsibilities. Fulfilling a caring role during emerging adulthood has been associated with poorer mental health. This study adopts a person-oriented approach in exploring the caring experience in relation to mental health. The sample comprised 3273 students (*M*_age_ = 20.19 years, *SD*_age_ = 1.89; 80.35% female) who answered a series of questionnaires. A cluster analysis identified six patterns of caregiving activities in terms of their nature and extent: few caring activities; household chores; household and financial/practical help; emotional care; sibling care; and many caring activities with emotional and personal care. A series of analyses showed that caring activities differed according to age, gender, living arrangements, financial status, the ill/disabled relative being supported, and the relative’s type of illness/disability. A multivariate analysis of covariance showed that emerging adult students with patterns featuring household chores had better mental health than those with few caring activities. Overall, our findings indicate that having an ill/disabled relative can lead an individual to take on a wide range of caring responsibilities that may have varying consequences for emerging adult students’ mental health.

## 1. Introduction

Many young adult students face the illness/disability of a relative on a daily basis (see Belghith et al. [1], for example). This situation may lead them to take on caring responsibilities such as cleaning, cooking, or administering medication. Young adults are deemed to be carers as soon as they provide regular help and support in relation to their age and culture [2]. Young adult carers (YACs) have therefore been defined as people aged 18–25 years “who provide or intend to provide care, assistance or support to another family member on an unpaid basis. The person receiving care is often a parent but can be a sibling, grandparent, partner, own child or other relative who is disabled, has some chronic illness, mental health problem or other condition (including substance misuse) connected with a need for care, support or supervision” [3] (p. 6). According to Leu et al.’s classification of countries’ awareness and policy responses to young carers’ issues, France, like Belgium, Ireland, Finland, and the United States, is classified as an *emerging country* with growing public and expert awareness [4]. If some services are developed for children and adolescent young carers, there are currently no public services specifically targeting YACs, as they were mentioned in a national mobilization strategy for the first time only in 2019 [5]. An initial estimation put the prevalence of YACs in the student population at 19% [6].

YACs are in the developmental stage of emerging adulthood [7], which is marked by gradual autonomy but relative independence from social roles [8]. There is an expectation that emerging adults will take on any caring responsibilities that present themselves, in line with individual, societal, and familial norms [2,9]. It is regarded as a family obligation or a normal responsibility that reaffirms emerging adults’ membership in their families [10,11].

### 1.1. Caring Activities

An ill/disabled relative often needs assistance in taking medication [7] or has a range of health and/or social needs [3]. To determine how much help a young person provides on a daily basis, caring activities can be placed on a *caregiving continuum* ranging from *light* to *very heavy* [2]. The light end corresponds to a low level of caring and responsibilities, mostly concerning activities of daily living, some being instrumental. It is a *routine* level that is considered age- and culturally appropriate and pertains to most young people. By contrast, the very heavy end is a high level of caring and responsibility. Caring activities are substantial, regular, and meaningful, as they include considerable help with instrumental activities of daily living. It is a level of caring that is age- and culturally inappropriate, involving more than 50 h of care per week, and only a small number of young people shoulder this. YACs may move from one end of the continuum to the other if there is an increase or decrease in the amount, regularity, complexity, intimacy, or duration of caring activities. 

The literature identifies six major caring activities in which emerging adults confronted with the illness/disability of a relative may be involved: domestic chores, household management, financial/practical help, emotional care, personal care, and sibling care [3,9,10,12,13,14,15,16,17,18]. *Domestic chores* are activities such as cleaning, cooking, and washing dishes or clothes. *Household management* consists of activities undertaken to keep the household running, such as shopping, household repairs, and lifting heavy objects. *Financial/practical help* is comprised of activities related to financial management (e.g., bills, social benefits, banking) and which engage young people in practical adult responsibilities such as working part-time or interpreting. *Personal care* involves activities such as administering health care (e.g., giving medication, changing dressings) or helping someone to dress, undress, wash, or use the bathroom. *Emotional care* consists in providing company and emotional support and attending to the care recipient’s emotional and psychological well-being [14]. Finally, *sibling care* involves activities such as looking after brothers or sisters.

Compared to adolescents, emerging adults are more likely to perform additional tasks, such as attending medical appointments, talking with health or other service providers, helping with bills, and providing emotional support [9]. Nevertheless, compared to older carers, they are less likely to perform *personal care* [7]. The literature shows that *emotional care* becomes more important when young people who have served a caring role since childhood or adolescence reach emerging adulthood [9]. Indeed, in emerging adulthood, *emotional care* is regarded as a core issue [3]. For YAC students, emotional care and household chores (i.e., domestic chores and household management) are the activities most often performed [12]. YAC students may also take on instrumental parental roles for their siblings [12]. 

### 1.2. Caring Activities and Mental Health

Taking on a caring role during emerging adulthood has been associated with poorer physical and mental health as well as reduced education, employment, and future life prospects [2,9]. YACs, in particular, are at greater risk of experiencing poor mental health [9]. They may feel worried, stressed, anxious, depressed, angry, upset, lonely, or resigned [2,9,11,19,20,21,22]. YACs’ mental health may be affected by several factors related to their caring roles [9], including the nature and extent of their caring activities.

When emerging adults are confronted with the illness/disability of a relative, the extent of caring activities is positively correlated with the likelihood of mental issues [18]. Even caring for a small amount of time is a risk factor for mental health [20,21,23]. Taking on a caring role has therefore been associated with mental health issues, with higher rates of psychological distress, as well as reduced subjective well-being [19,20,21]. Moreover, YAC students appear more vulnerable than other students to psychological distress relative to the burden of caring [19] in terms of the extent and nature of caring activities [11]. Mental health issues have also been linked to the types of caring activities. YACs exhibit greater distress than others when they provide *personal care* only or in combination with *household management* [20]. In addition, *personal care* and *emotional care* have been identified as the responsibilities with the most significant negative impacts on youth [24]. 

### 1.3. The Present Study

In the literature, the caring experience has been investigated primarily through qualitative approaches using “small-scale studies” with a selected groups of carers despite the fact that young carers are not a homogeneous group [25]. To be considered more seriously by policy makers, Joseph et al. [25] have recommended developing large-scale quantitative research. Up to date, quantitative research focuses on the relationships between variables at the level of the entire sample, applying a variable-oriented approach. This approach allows authors to test the ways in which variables might relate to each other, though it cannot show the joint effect of variable combinations with more than two or three interacting predictors. It does not reflect the dynamic nature of individuals’ experiences [26] or, consequently, the heterogeneity within the group of young carers. In contrast, a person-oriented approach allows individual functioning to be understood as a dynamic system of interwoven components so that subgroups of participants characterized by similar life experiences can be identified. When the caring experience in examined from a dynamic perspective, it becomes possible to investigate the different configurations that various caring activities can take across the range of individuals and not just one caring activity by one. Above all, for our purposes, it enables us to grasp the specificity of the caring experience and its implications for mental health issues.

This study applies a person-oriented approach to examine the diversity of caring experiences through caring activities and their links to mental health for emerging adult students. First, we investigated the various forms of caring activities. We expected several patterns to emerge, underlining the heterogeneity of experiences of caring activities. In line with the literature on YAC students [12], we predicted that one pattern would be characterized by emotional care, one by household chores (i.e., domestic chores and household management), and another by sibling care. Given the continuum of caring activities [2], a fourth pattern would be observed at the light end of the continuum, with a low level of caring activities, and a fifth at the very heavy end, with a high level of caring activities including personal care. These contrasting patterns would reflect the diversity of the caring activities taken on by emerging adults. 

Second, adopting an exploratory perspective and aiming to assess the caring experience specificity of each pattern, we expected these patterns to be linked to age, gender, living arrangements and financial status, as well as the ill/disabled relative’s identity and type (s) of illness/disability. Third, we analyzed the relationships between caring activity patterns and mental health (i.e., psychological distress and subjective well-being). Based on the literature [20,24], we assumed that patterns characterized by lighter caring activities would be associated with lower levels of psychological distress and higher subjective well-being in contrast to those characterized by emotional care, household chores, or multiple caring activities, including personal care. Furthermore, participants with patterns characterized by emotional care and personal care would display more mental health issues than others.

## 2. Materials and Methods

### 2.1. Participants

The present study was conducted as part of the CAMPUS-CARE research project, which investigated emerging adult students’ health and well-being. A total of 9571 emerging adult students agreed to participate by completing a consent form. The criteria for inclusion were being 18–25 years of age, having no children, and being enrolled in a higher education program. For the purposes of this study, participants with missing responses to the Multidimensional Assessment of Caring Activities for Young Carers (MACA-YC18; 18,16 for the French version) were excluded (*n* = 2804), as were participants who did not report being confronted with the illness/disability of a relative (*n* = 3494). The final sample therefore was comprised of 3273 emerging adult students (*M*_age_ = 20.19, *SD*_age_ = 1.89; 80.35% females). Participants were enrolled in a range of courses at the undergraduate or graduate level (43.51% literature, art, and human sciences; 17.87% law and economics; 16.74% sciences and technology; 16.01% medicine; 3.21% engineering; 1.25% education). Concerning the ill/disabled relative, 61.29% of participants stated that they had two or more such relatives, and 42.19% of those participants stated that they lived with one of them. For each of 38.09% of participants, their mother was (one of) the ill/disabled relative (s); for 33.85%, their father; for 21.57%, a sibling; for 26.66%, a grandparent; and for 56.82%, some other relative. These relatives had either a physical illness (e.g., cancer, diabetes; 31.90%), a mental illness or substance use disorder (e.g., depression, eating disorder; 40.78%), a disability (e.g., paraplegia; 15.17%) or other illnesses/disabilities (27.32%). Participants’ characteristics are set out in Table 1. Data were provided on a voluntary basis through an anonymous self-report questionnaire.

### 2.2. Measures

*Sociodemographic information*. Several questions assessed age, gender, education level, higher education program, siblings, living arrangements, financial status, and student employment.

*Illness/disability of a relative*. Participants were asked whether a friend or family member suffered from a chronic physical illness, mental illness or substance abuse, disability, or some other health issue. If they answered “yes”, they then had to indicate who had this issue and whether they lived with that person.

*Perception of support provided*. The About Me and My Family [15], ([27] for the French adaptation) assessed whether participants perceived themselves as providing regular support for a relative: “Some people have a family member or friend who needs support on a regular basis, for example, because he/she is ill or disabled. Do you support someone on a regular basis?”. An affirmative answer was followed up with questions about who the relative (s) was (were) and the reason for providing support. To determine whether or not the support was provided due to the relative’s illness/disability, two researchers independently analyzed the reasons given for providing this support. Inter-rater agreement was satisfactory based on the usual criteria (i.e., κ = 0.77, *p* < 0.001; [28]). Differences in ratings were discussed until a consensus was reached.

*Caring activities*. Caring activities were assessed with the French version of the MACA-YC18. This 18-item questionnaire evaluates the six caring activity domains: domestic chores, household management, financial/practical help, emotional care, personal care, and sibling care. Items are rated on a scale ranging from 0 (“Does not concern me/Never”) to 2 (“A lot of time”). Cronbach’s alphas, reported in Table 2, were similar to those for both the original version [15] and the French version [13].

*Mental health*. Psychological distress was assessed with the French version [29] of the 12-item General Health Questionnaire (GHQ-12) [30]. Respondents were asked to indicate the frequency with which they had experienced symptoms of mental health issues over the previous few weeks using the Likert method (all items coded 0-1-2-3). The total score ranges from 0 to 36, with a higher score indicating greater psychological distress. In the general population, a score above 12 is indicative of symptoms of depression [31,32]. The Cronbach’s alpha for our sample was satisfactory (α = 0.92).

*Subjective well-being* was assessed with the Satisfaction With Life Scale (SWLS) [33], ([34] for the French version). This questionnaire contains five items rated on a 7-point Likert scale. The total score varies between 5 and 35, with a higher score indicating greater life satisfaction. Cronbach’s alpha for our sample was satisfactory (α = 0.84). 

### 2.3. Plan of Analysis

First, to investigate patterns of caring activities, we ran a two-step cluster analysis. In the first step, we performed a hierarchical cluster analysis using Ward’s method and squared Euclidean distance. This yielded initial cluster centers which, in the second step, were taken to constitute a nonrandom starting point in an iterative *k*-means analysis. The final cluster selection was based on three criteria, e.g., [35]: substantive theorizing, parsimony, and explanatory power (i.e., the greatest variance explained in each dimension). We used conventional criteria to interpret the clusters [36]: an absolute value of 0.2 *SD* defined a small effect; 0.5 *SD* a moderate effect; and 0.8 *SD* a large effect. As there were no missing data, the analysis was run on the entire sample.

Second, to characterize the patterns of caring activities, we ran an analysis of variance (ANOVA) and chi-square tests. Patterns were characterized according to participants’ age, gender (male, female, other), whether they lived with one or both parents, financial status (better than others, same as others, worse than others), whether they lived with an ill/disabled relative, the identity of the ill/disabled relative they supported (mother, father, sibling, grandparent, other), and the relative’s type of illness/disability (physical illness, mental illness, disability, other illnesses/disabilities, several illnesses/disabilities). The ANOVA was supplemented with a Tukey post hoc test using pairwise comparisons as well as a chi-square test examining the standardized residuals (i.e., an absolute value above 2 indicated which cells differed significantly from the hypothesis of independence). As there were no missing data, the analysis was run on the entire sample.

Third, to determine whether there were any associations between caring activity patterns and mental health variables, we performed a multivariate analysis of covariance (MANCOVA), controlling for gender with Tukey’s post hoc pairwise comparisons. Once we established that there were no outliers, the analysis was run on the entire sample. Analyses were conducted with the stats 4.1.2, rrcov 1.6-2, MASS 7.3-55, ggplot2 3.3.5, dplyr 1.0.7, and psych 2.1.9 packages in R 4.1.2 software [37]. Our analysis plan is reported in the flowchart (see Figure 1).

## 3. Results

### 3.1. Descriptive Statistics

All descriptive statistics and correlations between caring activity domains and mental health are reported in Table 2. Caring activity domains were significantly intercorrelated (0.41 > *r* > 0.10), as were mental health variables (0.79 > *r* > −0.52). However, caring activity domains were only weakly correlated with mental health. Psychological distress correlated negatively with domestic chores (*r* = −0.08) and positively with emotional care (*r* = 0.07). Subjective well-being was positively correlated with domestic chores (*r* = 0.09) and negatively correlated with financial/practical help (*r* = −0.07) and sibling care (*r* = −0.06).

### 3.2. Patterns of Caring Activities

Combining the six caring activity domains, the cluster analysis yielded a six-cluster solution (see Figure 2). This cluster solution accounted for 25.40% of the variance in domestic chores, 42.58% of the variance in household management, 51.38% of the variance in financial/practical help, 57.08% of the variance in emotional care, 73.14% of the variance in personal care, and 62.56% of the variance in sibling care. A discriminant function analysis supported this final cluster solution: Wilks’ lambda = 0.02, χ^2^ (30) = 12606, *p* < 0.001; 95.78% of cross-validated grouped cases correctly classified. 

There were six patterns of caring activities. Participants in the *few caring activities* cluster scored low or very low in all domains. The *household chores* cluster had moderately high scores on domestic chores and household management and low or very low scores on the other domains. The majority of our sample (55.93%) belonged to these two clusters. The *household and financial/practical help* cluster was characterized by very high scores on financial/practical help, high scores on household management and domestic chores, and intermediate scores in the other domains. The *emotional care* cluster had intermediate scores in all domains except emotional care, where it had very high scores. The *sibling care* cluster had very high scores on sibling care and moderately high scores in the other domains. Finally, the *many caring activities with emotional and personal care* cluster had high scores in all domains and very high scores on emotional and personal care; 5.25% of our sample belonged to this pattern.

### 3.3. Characterization of Patterns of Caring Activities

Caring activities clusters differed significantly within the following: age, *F* (5) = 23.00, *p* < 0.001, η^2^ = 0.03; gender, χ^2^ (*N* = 3273, *df* = 10) = 22.84, *p* < 0.05; living with one or both parents, χ^2^ (*N* = 3273, *df* = 5) = 92.99, *p* < 0.001; financial status, χ^2^ (*N* = 3273, *df* = 5) = 111.97, *p* < 0.001; student employment, χ^2^ (*N* = 3273, *df* = 10) = 224.13, *p* < 0.001; living with an ill/disabled relative, χ^2^ (*N* = 3273, *df* = 5) = 29.50, *p* < 0.001; identity of the ill/disabled relative being supported, χ^2^ (*N* = 3273, *df* = 5) > 65, *p* < 0.001; and relative’s type of illness/disability, χ^2^ (*N* = 3273, *df* = 5) > 20, *p* < 0.001. No significant differences obtained for the categories of other relatives (χ^2^ (*N* = 3273, *df* = 5) = 3.81, *p* = 0.58) or other illnesses/disabilities (χ^2^ (*N* = 3273, *df* = 5) = 6.82, *p* = 0.23). Results are reported in Table 3. 

The *few caring activities* pattern was mostly represented by males living with one or both parents, with a better perceived financial status, and no student job. It was least represented by students supporting a mother, father, sibling, or grandparent with either one physical illness or disability or with several illnesses/disabilities. The *household chores* pattern was mostly represented by students with better perceived financial status and no student job and least represented by students living with one or both parents; living with an ill/disabled relative; supporting a mother, father, sibling, or grandparent; having relatives with a mental illness; and supporting relatives with several illnesses/disabilities. The *household and financial/practical help* pattern was mostly represented by students who were the oldest sibling, had a worse perceived financial status, frequently or occasionally had a student job, supported an ill/disabled mother or father, supported relatives with a physical illness, and supported relatives with several illnesses/disabilities. It was least represented by students who lived with one or both parents. The *emotional care* pattern was mostly represented by students who were living with an ill/disabled relative, supported an ill/disabled mother, father or grandparent, supported relatives with a physical or mental illness, and had relatives with several illnesses/disabilities. The *sibling care* pattern was mostly represented by female students who were the youngest sibling; lived with one or both parents; lived with an ill/disabled relative; had a worse perceived financial status; did not have a student job; supported an ill/disabled mother, father, or sibling; and had relatives with a physical illness. It was least represented by students who had relatives with a mental illness. Finally, the *many caring activities with emotional and personal care* pattern was mostly represented by female students who occasionally had a student job; supported an ill/disabled mother, sibling, or grandparent; had relatives with a disability; and had relatives with several illnesses/disabilities. It was least represented by students with a better perceived financial status.

### 3.4. Relationship between Patterns of Caring Activities and Mental Health

The MANCOVA revealed a significant effect of caring activity pattern on mental health after controlling for gender, *F* (10, 6510) = 3.74, *p* < 0.001. There were significant effects on psychological distress in terms of the GHQ-12 (*F* (5, 183,277) = 3.09, *p* < 0.01, η^2^ = 0.025) as well as subjective well-being, per the SWLS (*F* (5, 153,069) = 4.59, *p* < 0.001, η^2^ = 0.016). Results are reported in Table 4. 

Tukey post hoc tests revealed that participants with the *household chores* pattern scored lower on psychological distress than those with the *few caring activities* or *emotional care* pattern. Participants with the *household and financial/practical help*, *sibling care*, or *many caring activities with emotional and personal care* pattern did not differ from the others. 

Finally, participants with the *household chores* pattern scored higher on subjective well-being than those with the *few caring activities*, *sibling care*, or *household and financial/practical help* pattern. Participants with the *emotional care* or *many caring activities with emotional and personal care* pattern did not differ from the others. 

## 4. Discussion

Using a person-oriented approach, the purpose of the present study is to gain insights into the caring experiences in relation to mental health among emerging adult students. We have highlighted the diversity of patterns of caring activities in terms of activities relating to domestic chores, household management, financial/practical help, emotional care, personal care, and sibling care. Analysis revealed a meaningful cluster solution based on the nature or extent of the caring activities. These patterns appear to be linked to age, gender, living arrangements, financial status, identity of the ill/disabled relative being supported, and the relative’s type of illness/disability. Our findings indicate that emerging adult students with patterns involving household chores had better mental health than those with few caring activities. Overall, our results showed that being confronted with an ill/disabled relative can lead young people to take on a wide range of caring responsibilities, and emerging adult students’ mental health should be considered in the light of their potential caring activities.

### 4.1. Patterns of Caring Activities

As expected, our findings highlighted various patterns of caring activities: few caring activities, household chores, household and financial/practical help, emotional care, sibling care, and many caring activities involving emotional and personal care. Four patterns were determined by the nature of the caring activities (i.e., *household chores*, *household and financial/practical help*, *emotional care*, *sibling care*) and two by their extent (i.e., *few caring activities*, *many caring activities with emotional and personal care*). Considering caring activities as diverse and numerous and adopting a person-oriented approach can therefore enable researchers to gain a finer perspective on emerging adult students’ caring experiences.

In line with Boumand and Dorant’s study [12], our results highlighted one pattern characterized by *household chores* (i.e., domestic chores and household management activities), one characterized by *emotional care*, and one characterized by *sibling care*. This finding reinforces the existing literature on emerging adult students, as it shows that emotional care is a core issue [3] and that emerging adult students are involved in household chores [12] and may take on instrumental parental roles for their siblings [12,38]. It is worth noting that these caring activity patterns pertained to approximately 50% of our sample. Thus, although they are important for emerging adult students, caring experiences should not be construed as being restricted to these three caring activities. 

Our results also revealed a pattern (*household and financial/practical help*) that was quite similar to *household chores* but with a higher degree of financial/practical help. This pattern concerned participants who were the oldest sibling, who had left their family households, who had a worse perceived financial status, and who frequently or occasionally had a student employment. The fact that one pattern was characterized by financial/practical help shows that emerging adults may also be involved in activities such as shopping for groceries, employment, managing finance, and arranging services [7,19]. We can assume that, as emerging adults can take on more responsibilities than adolescents [8], they are more involved in financial and practical help than younger carers. There may also be a greater social expectation on them to take on such responsibilities as they become more autonomous [3]. The *household and financial/practical help* pattern may therefore be an expression of emerging adult students’ developmental stage of life rather than of ill/disabled relatives’ needs. This underlines the importance of considering YACs in terms of their developmental stages and related societal expectations both in research and in practice. 

Finally, as expected, our results revealed two extreme patterns: *few caring activities* and *many caring activities with emotional and personal care*. The former can be placed at the light end of Becker’s [2] continuum of caring activities, and the latter can be placed at the very heavy end. The *few caring activities* pattern shows that emerging adult students can be confronted with the illness/disability of a relative without necessarily taking on any caring responsibilities. This supports the notion that the extent of caring activities can be used to differentiate young carers from non-carers [2,39]. Although there is no single procedure for identifying YACs [40], our findings suggest that, in alignment with recommendations [13,16,40], the extent of caring activities should be used as an identification criterion. Developing a single procedure would improve the identification of YACs as well as the support they are given [25] and would help avoiding the problem of hidden carers [41]. It is worth noting that the *few caring activities* cluster contained the most participants (28.21% of the sample), and only 5.25% of the sample belonged to the *many caring activities with emotional and personal care* cluster. This particular finding indicates that only a very small number of emerging adult students take on high levels of caring responsibilities including personal care activities [7].

### 4.2. Characteristics of Caring Activity Patterns

Our analysis of pattern characteristics revealed several differences. The oldest emerging adult students tended to be in the *household and financial/practical help* cluster, whereas the youngest were in the *sibling care* cluster. This finding reinforces the idea that with age, emerging adult students become more independent [8] and take on more adult-like responsibilities (e.g., employment), especially if they perceive their financial status to be worse than that of others. We can assume that the emerging adult students exhibiting this pattern were the more independent ones, as they had also left their family households. Nevertheless, although personal care is more likely to be provided by older carers [7], emerging adult students with the *many caring activities with emotional and personal care* pattern were not the oldest participants. This result indicates that personal care activity is more dependent on the caring situation than on the development of the emerging adult student. Regarding the *sibling care* pattern, these emerging adult students were less independent, as they lived with one or both of their parents. They also were not employed even when they perceived their financial status to be worse than that of others. The process of gaining independence in emerging adulthood may therefore have an impact on caring responsibilities. YACs should thus be considered throughout the emerging adulthood developmental stage of life.

Gender bias was confirmed, as more females than males engaged in *sibling care* and in *many caring activities*, including personal care. Males appeared to engage less in caring activities. These findings are similar to those of Becker and Sempik [42] for youth 14–25 years of age, and they show that, even now, society continues to assign to females the role of caring for and helping others [43]. The common characteristic of these two patterns was an overrepresentation of ill/disabled mothers and siblings. Emerging adult students may feel high levels of obligation toward their mothers, especially if they are female [44]. Within a sibship, sisters are more likely than brothers to take on caring responsibilities and spend more time on caring activities [42,45]. Gender bias may therefore be related to social expectations concerning both caring activities and the ill/disabled relatives who are supported.

No type of relative was overrepresented in the *few caring activities* and *household chores* patterns, emphasizing that emerging adults with these patterns may not take on a caring role or may support the whole family through instrumental activities. We can assume either that the ill/disabled relative did not need a high degree of help or that emerging adults with these patterns were not the primary carers but rather the secondary or tertiary carer. Although YACs are mostly primary carers [46], YAC students may be secondary or tertiary ones [11] and may take on a fewer level of caring responsibilities [47] or instrumental activities [48]. This assumption is reinforced by the fact that emerging adult students with the *few caring activities* pattern lived in their family households, whereas those with the *household chores* pattern had left their family households and did not live with ill/disabled relatives. They could not, therefore, be primary carers. 

Furthermore, it is worth noting that emerging adult students with the above patterns had a better perceived financial status than others and were not employed. Poor financial circumstances are related to performing a caring role [3]. We can assume that a good perceived financial status is a protective factor against caring responsibilities. In contrast, providing support to an ill/disabled mother requires engagement in a wide range of caring activities, including household help, financial/practical help, emotional care, sibling care, and personal care. There were similar findings for emerging adult students who supported an ill/disabled father except that they did not engage in many activities involving emotional and personal care. These findings are congruent with the literature, as caring activities are broader and more extensive when the care recipient is a mother rather than a father, in line with social expectations [49,50]. 

Furthermore, providing support to an ill/disabled sibling or grandparent involves engaging in many caring activities, including providing emotional and personal care. Emotional care seemed to be more important when the relative was a grandparent, and sibling care seemed to be more important when the relative was a sibling. Emotional care can be a care-at-distance task that can be performed even if the emerging adults do not share accommodation with the care recipient [3], as is very often the case in France with grandparents (see Belghith et al., [51]). Nevertheless, emerging adult students with this pattern mostly lived with an ill/disabled relative, confirming that emotional care is still a core issue within households. Above all, emerging adults could help parents as well as siblings and grandparents. However, the nature and extent of their caring activities seemed to depend on which ill/disabled relative was being supported. 

Finally, regarding the relative’s type of illness/disability, our results showed that the physical illness of a relative played a large role in determining whether participants would engage in household, financial/practical, emotional, or sibling care. In cases where the relative had a mental illness, emotional care was the more important activity, whereas a physical disability entailed many caring activities, including emotional and personal care. These findings are consistent with the literature: being confronted with a relative who has a physical illness or disability leads to high scores on domestic chores and emotional care activities [49,50], whereas emotional care is more frequent than other activities when the relative has a mental illness [52]. Moreover, emerging adult students whose relatives had several illnesses/disabilities engaged in many caring activities, including household chores, financial/practical help, and emotional and personal care, regardless of whether they lived with the ill/disabled relative. Complementing our previous findings, these results show that both the nature and the extent of caring activities depend on the relatives’ needs in terms of their illnesses/disabilities [49,53,54]. Overall, our findings raise questions regarding the ways in which emerging adult students become caregivers and their willingness to take on caring responsibilities. These questions seem especially important, as the literature shows that some emerging adults take on these responsibilities because there is no other option [14], while for others, it is a personal choice that they freely make [23]. It is therefore legitimate to ask how the entry into caring might influences the experience of caring and how it might affect caring’s consequences for mental health.

### 4.3. Patterns of Caring Activities and Mental Health

Patterns of caring activities were differently associated with mental health (i.e., psychological distress and subjective well-being). Contrary to the literature [20,24] and our expectations, emerging adult students in the *household chores* cluster had the best mental health, whereas those in the *few caring activities* cluster had the worst mental health. Although providing personal care was associated with greater distress [20] and the need for support [16], emerging adult students in the *many caring activities with emotional and personal care* cluster had neither better nor worse mental health. As regards psychological distress, it is worth noting that nearly all participants had high GHQ-12 scores, suggesting a high rate of depressive symptoms [31,32]. The literature shows that providing support has roughly the same impact on psychological distress regardless of the amount of time it takes [20]. We can assume that, regardless of the amount of caring, the simple fact of being confronted with an ill/disabled relative increases the risk factor of mental health problems. 

Furthermore, the fact that participants with the *few caring activities* pattern had poorer mental health than those with the *household chores* pattern shows that taking on instrumental caring activities may be a way of trying to cope with the ill/disabled relative’s situation [50]. More specifically, taking on caring responsibilities may lead to a sense of control over the situation [50,55], which is something young people struggle to maintain in the face of uncertainty [56]. This explanation is reinforced by the fact that emerging adult students in the *household chores* cluster, which involves instrumental activities of daily living, had the best mental health. The feeling of a lack of control is associated with negative outcomes such as stress [57]. We can assume that emerging adult students in the *few caring activities* cluster had precisely this feeling of lack of control. The nature of the caring activity, the relative’s type of illness/disability, and the emerging adult students’ mental health are thus interrelated. This finding indicates that a holistic perspective (e.g., person-oriented approach), which considers the individual as a whole [26], better captures emerging adult students’ caring experiences and therefore facilitates more tailored support. 

As for the other patterns based on the nature of caring activities, emerging adult students seemed to have specific mental health issues. Those with the *household chores and financial/practical help* pattern scored lower on subjective well-being than those with the *household chores* pattern. They took on student jobs and reported a worse perceived financial status than others. *Financial/practical help activities* are, for our purposes, construed as activities that engage emerging adults in additional practical responsibilities, such as part-time employment. Having a student job is associated with a heightened feeling of being overwhelmed [58], which can influence well-being [59]. Our findings indicate that taking on a financial role with a student job in addition to household tasks is a risk factor for diminished well-being. Furthermore, perceived financial status is linked to well-being [60]. Emerging adult students with the *emotional care* pattern scored higher on psychological distress than those with the *household chores* pattern. Although domestic chores and household management are time-consuming instrumental activities [61], they require less psychological investment than emotional care. The latter is perceived to be difficult, as it may, among other things, restrict participation in social life events [3]. Peer support, which is favored during social life events, is a protective factor against depressive symptoms [62]. 

Our findings also showed that emerging adult students with the *emotional care* pattern tented to live with an ill/disabled relative and, in most cases, were confronted with relatives who had several illnesses/disabilities. The emotional care items of the MACA-YC18 focus on companionship, vigilance, and accompaniment [40]. We can assume that these kinds of activities are easier for emerging adult students who live with the ill/disabled relative. In addition, caring for a relative with comorbidities brings a risk of emotional difficulties [63] and is associated with reduced social connectedness [64]. The higher psychological distress score for emerging adult students with the *emotional care* pattern can therefore be explained by both the nature of the caring activity, the living arrangements, and the relative’s type of illness/disability. Finally, emerging adult students with the *sibling care* pattern had a lower subjective well-being score than those with the *household chores* pattern. They lived with one or both parents as well as with an ill/disabled relative, who was the mother, father, or sibling. Sibling care activities may involve playing an instrumental parental role in the household [12,38]. Parentification, an instrumental parental role, has negative consequences for young carers [65]. All in all, we can reasonably conclude that roles undertaken when confronted with the illness/disability of a relative influence emerging adult students’ mental health. 

### 4.4. Limitations and Suggestions for Future Research

The strength of our person-oriented study is that its findings underscore the diversity of caring experiences in terms of the nature and extent of the caring activities. It provides a better understanding of how these activities can undermine the mental health of emerging adult students. Nevertheless, as in both the original version of the MACA-YC18 [15] and the French version [13], three of the dimensions we examined (domestic chores, household management, financial/practical help) exhibited poor reliability. This lack of reliability may be explained by the fact that these dimensions cover a broad range of activities and are therefore conceptualized as having several distinct aspects [66]. Moreover, because emerging adult students may be secondary or tertiary caregivers [11], and because the amount of caring decreases as soon as caring responsibilities are shared [3], we can assume that caring activities may differ according to whether the individual is the primary caregiver or the secondary or tertiary caregiver. In addition, one-third of our participants provided support to ill/disabled grandparents. In an intergenerational perspective, it has been shown that the more parents care for a grandparent, the more likely it is that the young will care for them as well [67]. If the emerging adult student is not the primary caregiver, it can be inferred that the relation between the primary caregiver and the care recipient will influence the extent and nature of the emerging adult students’ caring activities. 

Aside from this, our study did not account for the number of emerging adult students’ friends or relatives supporting them. This is relevant, as close friendships and family cohesion contribute to young carers’ social capital [68], and young people confronted with an ill parent expressed the need to feel less alone [69]. Knowing that loneliness may influence mental health [70], we can suppose that the amount of perceived social support can moderate the link between caring activities and mental health issues. Furthermore, given that the extent of caring responsibilities may increase or decrease over time, depending on the relative’s state of health [3], adopting a longitudinal perspective would allow researchers to observe changes in emerging adult students’ caring experiences and the implications for their mental health. In addition, our findings raise questions about emerging adult students’ willingness to take on caring activities [14,23]. Assessing the feeling of control related to caring experiences could further refine our perspectives on these students’ mental health. 

Finally, our study was partly conducted during the COVID-19 pandemic. YACs are more vulnerable than non-carers to pandemic-related distress and mental health problems [71]. The pandemic and its influence on emerging adults’ mental health should be considered in future research so as to better understand the links between the caring experiences and mental health issues. Overall, further in-depth examinations of the caring phenomena would help to fine-tune our present results in order to better support emerging adult students confronted with ill/disabled relatives.

## 5. Implications

These findings may have significant implications for supporting emerging adult students confronted with the illness/disability of a relative and for preventing and addressing their mental health issues. Our results illustrate the importance of considering YACs in terms of their developmental stage in relation to societal expectations. Most services for young carers are not relevant to people over 18 years old [3]. Furthermore, adult support services are not adequately matched with YACs needs, as they are generally advocated for and used by older carers [72]. Developing dedicated support services without transposing existing ones and while also adapting them to the specifics of YACs’ circumstances is a challenge. In discussing the university context, Kettell [73] proposed several solutions, such as promoting the establishment of YAC student associations or developing peer mentors during the first semester or a “carers’ passport”, which grants priority access. 

Moreover, our findings highlight that emerging adult students in the *few caring activities* pattern had the poorest mental health compared to others. Dedicated services within universities should thus be accessible to all emerging adult students confronted with the illness/disability of a relative and not only to those who have identified themselves as YACs. Those services should then consider emerging adult students who “care about” a relative and those who “care for” a relative [2]. Furthermore, our results suggest that taking on the instrumental activity pertaining to of daily living may lead to a sense of control over the situation [50,55] and, consequently, to improved mental health. To foster this feeling of control, it could be imagined as a space for talking, which offers the opportunity to communicate during and around the caring situations [17]. Moreover, by investigating the caring experience in a person-oriented approach, our findings underlined that the extent of caring activities should be considered for YACs’ identification [13,16,40]. Overall, as young carers are not a homogeneous group [25], identifying the caring activity patterns, their characteristics, and their link with mental health issues is a first step to better capture the diversity of caring experience.

## 6. Conclusions

This study adopted a person-oriented approach to better understand emerging adult students’ caring activities, their characteristics, and the corresponding effects on mental health. Our findings emphasize the diversity of caring experiences, underscoring the importance of considering variations in both the nature and the extent of caring activities. Our results also highlighted that emerging adult students should not be construed as being restricted to household chores, emotional care, or sibling care activities. Furthermore, the extent of caring activities should be used in procedures for identifying YACs. Moreover, taking on instrumental activities was founded to be related to better mental health, as it may provide a feeling of control over a situation of considerable uncertainty without requiring a high degree of psychological investment. Future research on student YACs should consider the feeling of control in relation to caring experiences as well as the willingness to take on caring responsibilities. Above all, future research as well as policy development should consider YACs in terms of their developmental stage and the related societal expectations.

## Figures and Tables

**Figure 1 ijerph-19-08104-f001:**
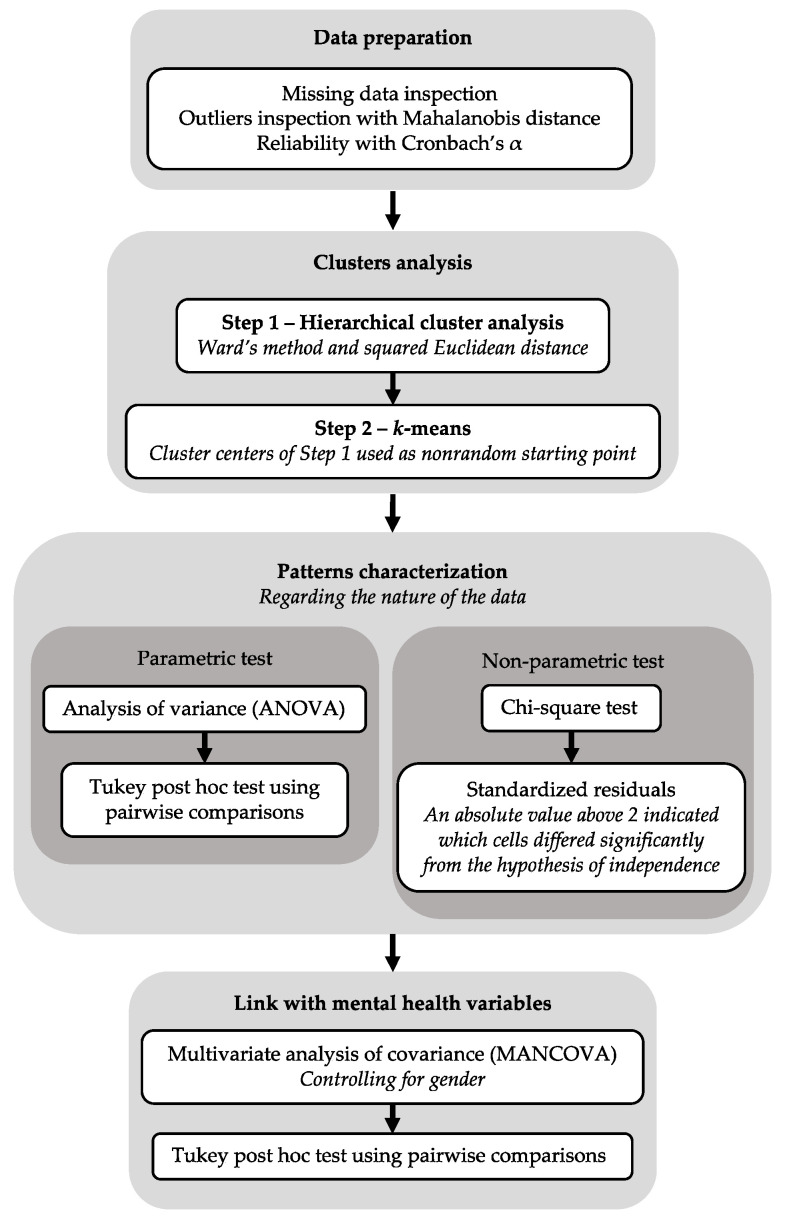
Flowchart of analysis plan.

**Figure 2 ijerph-19-08104-f002:**
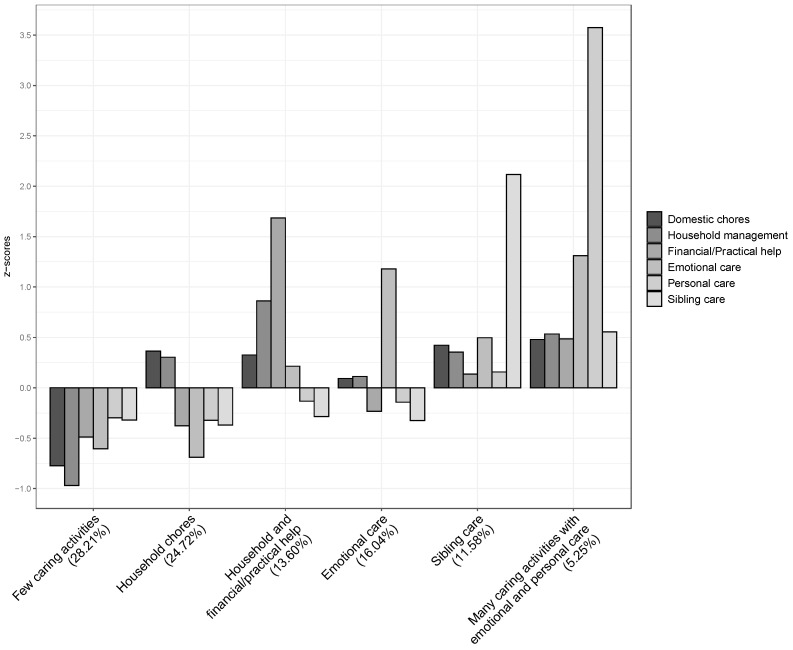
Cluster solution for caring activities. *N* = 3273. z-scores for domestic chores, household management, financial/practical help, emotional care, personal care, and sibling care.

**Table 1 ijerph-19-08104-t001:** Sample sociodemographic characteristics (*N* = 3273).

	Total Sample
Gender: females, *n* (%)	2630 (80.35)
Mean age in years (*SD*)	20.19 (1.89)
Grade, *n* (%)	
First year	1141 (34.86)
Second year	828 (25.30)
Third year	621 (18.97)
Fourth year	370 (11.30)
Fifth year	234 (7.15)
Sixth year	79 (2.41)
Program, *n* (%)	
Literature, art, and human sciences	1424 (43.51)
Law and economics	585 (17.87)
Sciences and technology	548 (16.74)
Medicine	524 (16.01)
Engineering sciences	105 (3.21)
Education and teaching	41 (1.25)
At least one sibling, *n* (%)	2931 (89.00)
Living with one or both parents, *n* (%)	1659 (50.70)
Perceived financial status, *n* (%)	
Better than others	925 (28.26)
Same as others	1933 (59.06)
Worse than others	415 (12.68)
Student employment, *n* (%)	
Yes, frequently	578 (17.65)
Yes, occasionally	705 (21.55)
No	1989 (60.79)
Ill/disabled relative, *n* (%)	
Mother	1247 (38.09)
Father	1108 (33.85)
Sibling	706 (21.57)
Grandparent	938 (26.66)
Other	1860 (56.82)
Relative’s type of illness/disability, *n* (%)	
Physical illness	1044 (31.90)
Mental illness/substance use disorder	1335 (40.78)
Disability	497 (15.17)
Other illnesses and disabilities	894 (27.32)
Relatives with several illnesses/disabilities, *n* (%)	1712 (52.31)
Living with an ill/disabled relative, *n* (%)	1381 (42.19)
Two or more ill/disabled relatives, *n* (%)	2006 (61.29)
Providing support to an ill/disabled relative, *n* (%)	1381 (42.19)
Mother, *n* (%)	1141 (76.83)
Father, *n* (%)	317 (21.34)
Sibling, *n* (%)	315 (21.21)
Grandparent, *n* (%)	400 (26.91)
Other, *n* (%)	538 (36.35)

**Table 2 ijerph-19-08104-t002:** Descriptive statistics, internal consistencies, and correlations between caring activity domains and mental health.

Variables	Min	Max	*M*	*SD*	α	2	3	4	5	6	7	8
Caring activities												
1. Domestic chores	0	6	3.72	1.22	0.59	0.31 ***	0.15 ***	0.17 ***	0.14 ***	0.14 ***	−0.08 ***	0.09 ***
2. Household management	0	6	2.48	1.47	0.49		0.37 ***	0.27 ***	0.16 ***	0.14 ***	−0.02	−0.01
3. Financial/practical help	0	6	1.20	1.39	0.59			0.25 ***	0.17 ***	0.11 ***	0.04	−0.07 ***
4. Emotional care	0	6	1.96	1.96	0.95				0.40 ***	0.23 ***	0.07 ***	−0.04
5. Personal care	0	6	0.44	1.18	0.89					0.24 ***	0.01	−0.04
6. Sibling care	0	6	0.98	1.67	0.94						0.03	−0.06 **
Mental health												
7. GHQ-12	0	36	19.66	7.58	0.92							−0.48 ***
8. SWLS	5	35	21.23	6.89	0.84							

*Note*. α, Cronbach’s alpha. ** *p* < 0.01. *** *p* < 0.001. GHQ-12, General Health Questionnaire; SWLS, Satisfaction With Life Scale.

**Table 3 ijerph-19-08104-t003:** Distribution of participants regarding age, gender, living arrangements, financial status, ill/disabled relative being supported, and relative’s type of illness/disability across caring activity patterns.

	Caring Activity Patterns						*F*/χ^2^	*df*	*p*
	Pattern A	Pattern B	Pattern C	Pattern D	Pattern E	Pattern F			
Age, *M (SD)*	20.13 ^b^ (1.89)	20.17 ^b^ (1.86)	20.91 ^a^ (2.02)	20.24 ^b^ (1.87)	19.56 ^c^ (1.68)	19.92 ^bc^ (1.63)	23.00	5	***
Gender, % (ASR)							22.84	10	*
Male	20.78 (**2.30**)	19.41 (0.91)	18.65 (0.19)	18.29 (−0.02)	13.72 (**−2.47**)	9.30 (**−3.14**)			
Female	77.73 (**−2.40**)	79.23 (−0.93)	80.00 (−0.20)	80.57 (0.13)	85.75 (**2.81**)	88.37 (**2.71**)			
Other	1.48 (0.54)	1.36 (0.13)	1.35 (0.07)	1.14 (−0.37)	0.53 (−1.43)	2.33 (1.20)			
Living with one or both parents, % (ASR)							92.99	5	***
	15.86 (**3.16**)	11.09 (**−3.82**)	4.80 (**−6.96**)	8.65 (1.60)	7.52 (**5.88**)	2.78 (0.59)			
Living with an ill/disabled relative, % (ASR)							29.50	5	***
	11.73 (−1.08)	9.10(**−3.57**)	5.50 (−0.80)	7.45 (**2.17**)	5.96 (**3.88**)	2.44 (1.17)			
Perceived financial status, % (ASR)							111.97	10	***
Better than others	9.66 (**4.24**)	7.76 (**2.28**)	2.90 (**−3.48**)	4.79 (0.91)	2.17 (**−4.34**)	0.98 (**−2.89**)			
Same as others	16.50 (−1.32)	15.09 (1.33)	7.67 (−1.22)	9.13 (−1.07)	7.27 (1.57)	3.39 (1.50)			
Worse than others	2.66 (**−3.77**)	1.86 (**−5.06**)	3.02 (**6.53**)	2.11 (0.35)	2.14 (**3.60**)	0.89 (1.69)			
Student employment, % (ASR)							224.13	10	***
Yes, frequently	3.88 (**−4.00**)	3.55 (**−2.89)**	5.04 (**11.55**)	2.41 (−1.69)	1.74 (−1.43)	1.04 (0.74)			
Yes, occasionally	5.13 (**−3.30**)	4.80 (−1.71)	3.76 (**3.36**)	3.67 (0.82)	2.20 (−1.28)	1.99 (**5.32**)			
No	19.80 (**5.91**)	16.38 (**3.67**)	4.80 (**−11.86**)	9.93 (0.63)	7.64 (**2.19**)	2.23 (**−5.06**)			
Ill/disabled relative supported, % (ASR)									
Mother	22.27 (**−9.62**)	26.45 (**−5.78**)	46.29 (**5.44**)	51.81 (**8.89**)	42.74 (**3.42**)	44.77 (**2.80**)	200.85	5	***
Father	5.19 (**−5.52**)	6.42 (**−3.61**)	17.75 (**6.19**)	13.14 (**2.92**)	12.66 (**2.09**)	11.63 (0.88)	76.41	5	***
Sibling	6.47 (**−3.89**)	6.92 (**−3.00**)	10.11 (0.37)	10.09 (0.40)	17.15 (**5.29**)	20.34 (**4.90**)	65.27	5	***
Grandparent	5.94 (**−6.98**)	6.43 (**−5.80**)	13.93 (1.18)	23.24 (**8.41**)	14.51 (1.45)	30.81 (**7.65**)	177.9	5	***
Other	11.88 (1.30)	12.61 (0.90)	20.22 (−0.74)	24.00 (−0.24)	18.20 (−0.52)	22.67 (−1.05)	3.81	5	0.58
Type of illness/disability, % (ASR)									
Physical	49.10 (**−5.33**)	53.40 (−1.96)	63.14 (**3.10**)	63.05 (**3.37**)	61.21 (**2.02**)	61.63 (1.43)	46.55	5	***
Mental/ Substance use	60.55 (1.78)	53.40 (**−3.15**)	60.00 (0.85)	63.61 (**2.78**)	53.05 (**−2.14**)	56.98 (−0.32)	20.99	5	***
Disability	23.01 (**−3.11**)	25.96 (−0.62)	26.97 (0.09)	28.95 (1.22)	28.49 (0.79)	40.70 (**4.23**)	25.93	5	***
Other	22.48 (0.64)	19.03 (−2.16)	22.02 (0.15)	21.71 (−0.02)	23.22 (0.73)	26.74 (1.63)	6.82	5	0.23
Relatives with several illnesses/disabilities, % (ASR)							52.09	5	***
	49.31 (**−2.18**)	44.00 (**−5.45**)	59.33 (**3.19**)	59.24 (**3.47**)	55.41 (1.29)	61.63 (**2.51**)			

*Note.* For age, we conducted a pairwise comparison using Tukey post hoc tests. For age, means sharing a subscript did not differ at *p* < 0.05. ASR, adjusted standardized residuals. Bold ASR reflects a significant over- or underrepresentation. Pattern A, few caring activities; Pattern B, household chores; Pattern C, household and financial/practical help; Pattern D, emotional care; Pattern E, sibling care; Pattern F, many caring activities with emotional and personal care. * *p* < 0.05. *** *p* < 0.001.

**Table 4 ijerph-19-08104-t004:** Mental health according to caring activity pattern.

	Caring Activity Patterns						*F* (*df*)	*p*	Post Hoc Comparisons	η^2^
	Pattern A	Pattern B	Pattern C	Pattern D	Pattern E	Pattern F				
GHQ-12	20.02 (7.62)	18.81 (7.49)	19.72 (7.73)	20.25 (7.51)	20.09 (7.41)	19.87 (7.22)	3.09 (5)	**	AD > B	0.025
SWLS	20.83 (6.83)	21.98 (6.92)	20.59 (6.67)	21.75 (6.83)	20.59 (7.16)	20.55 (6.70)	4.59 (5)	***	B > AEC	0.016

*Note*. Standard deviations are in parentheses. Pattern A, few caring activities; Pattern B, household chores; Pattern C, household and financial/practical help; Pattern D, emotional care; Pattern E, sibling care; Pattern F, many caring activities with emotional and personal care. GHQ-12, General Health Questionnaire; SWLS, Satisfaction With Life Scale. ** *p* < 0.01. *** *p* < 0.001.

## Data Availability

The data presented in this study are available on request from the corresponding author.

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
