# Peer review of "Are We All the Same When Faced with an Ill Relative? A Person-Oriented Approach to Caring Activities and Mental Health in Emerging Adult Students"

_ijerph, 2022, doi:10.3390/ijerph19138104_

Round 1

Reviewer 1 Report

Thank you for the opportunity to read this manuscript and congratulations to the authors for their work.

Here are some suggestions for improvement:

Introduction: I miss data on the characteristics of care in the geographical area of the study (France). These data are necessary to understand the framework of the study and contextualize the results.

Method: Table 1 is unclear. I suggest you restructure the table. The description of the analysis carried out is very complete, and since it has been a complex analysis it is staggered, I suggest the elaboration of a flowchart that helps to visualize the global analysis.

Conclusions: they are poor compared to the discussion. 

Author Response

# Reviewer 1

Thank you for the opportunity to read this manuscript and congratulations to the authors for their work.

Here are some suggestions for improvement:

Introduction: I miss data on the characteristics of care in the geographical area of the study (France). These data are necessary to understand the framework of the study and contextualize the results.

Thank you for your comments. As requested, we have added information about the characteristics of care in France, as follows (cf. p.1, lines 37-42 and p.2, lines 43-44): “According to Leu et al.’s classification of countries’ awareness and policy responses to young carers’ issues, France, like Belgium, Ireland, Finland, and the United States, is classified as an emerging country with growing public and expert awareness [23]. If some services are developed for children and adolescents young carers, there are currently no public services specifically targeting YACs, as they were mentioned in a national mo-bilization strategy for the first time only in 2019 [24]. An initial estimation put the prevalence of YACs in the student population at 19% [25].”

Method: Table 1 is unclear. I suggest you restructure the table. The description of the analysis carried out is very complete, and since it has been a complex analysis it is staggered, I suggest the elaboration of a flowchart that helps to visualize the global analysis.

Table 1, which refers to the sociodemographic characteristics, has been formatted to improve clarity.

As recommended, a flowchart of the analysis plan has been added (cf. Fig. 1, p.6).

Conclusions: they are poor compared to the discussion. 

The conclusion section has been revised (cf. p.15, lines 607-609 and p.16, lines 610-621). In addition, in accordance with one of Reviewer 4’s suggestions, an Implications section has been added (cf. p.15, lines 579-606).

Reviewer 2 Report

I greatly appreciated having the opportunity to read this article. The performance of this study was unique, with a high reliance on questionnaires to obtain data (page 4). Since numerous factors may affect the mental health of young adult carers, analyzing other possible factors, such as the amount of friends or relatives that support the young adult carers or their willingness to provide care, may be beneficial to the study and strengthen the results.

Review "the aim of the present study" (page 3, lines 109-128), and describe both the program used for estatistical calculations and ethical considerations at the end of materials and methods section (line 211, page 6).

Edition: Please, review lines 36-37 (page 1), also the end of line 41, and format table 3 (page 8 and 9), as some numbers might be incorrectly understud (minus symbol in different lines). Also review the sentences from lines 30 ("...for explample see [1].") and line 329 ("In line with [10]'s study,..."]

Author Response

# Reviewer 2

I greatly appreciated having the opportunity to read this article. The performance of this study was unique, with a high reliance on questionnaires to obtain data (page 4). Since numerous factors may affect the mental health of young adult carers, analyzing other possible factors, such as the amount of friends or relatives that support the young adult carers or their willingness to provide care, may be beneficial to the study and strengthen the results.

Thank you for your comments. Unfortunately, we do not have information about the number of friends or relatives that support YACs. We agree that this factor may be beneficial. It is now discussed as a limitation of the study regarding Barry’s (2010) findings on young carers’ social capital and the findings of Kavanaugh et al. (2015) regarding young carers’ isolation (cf. p.15, lines 559-564).

Regarding the willingness to provide care, we already mentioned the motive to become carers as a new object of research that could strengthen our present results. To better clarify this point, we have revised this element (cf. p.13, lines 471-477 and p.15, lines 568-571).

Review "the aim of the present study" (page 3, lines 109-128), and describe both the program used for estatistical calculations and ethical considerations at the end of materials and methods section (line 211, page 6).

As requested, and also in accordance with a comment from Reviewer 4, the present study section has been revised (cf. p.3, lines 107-127).

The program used for our statistical calculations has been added at the end of the Plan of Analysis section (cf. p.6, lines 231-233). The ethical considerations are already mentioned in the Institutional Review Board Statement section (cf. p.14, lines 549-552) and have been added to the Participants section (cf. p.4, lines 150-151).

Edition: Please, review lines 36-37 (page 1), also the end of line 41, and format table 3 (page 8 and 9), as some numbers might be incorrectly understud (minus symbol in different lines). Also review the sentences from lines 30 ("...for explample see [1].") and line 329 ("In line with [10]'s study,..."]

Lines 36–37 (p. 1) have been reviewed by inserting the citation in the main text.

The end of line 41 was revised as ‘characterized’ was replaced by ‘marked’.

Table 3 has been reformatted to improve clarity (cf. pp. 9-10).

Line 30 has been rephrased as well as line 329 by inserting the authors’ names.

Reviewer 3 Report

I highly appreciate the text submitted for review.

Care for a sick, elderly or disabled person is often the responsibility of the family members of that person. Notably, state actions are not able to release society from this activity. I even think it would be inappropriate. That is why it is so important to analyze the situation of carers (mental health also). This research fits very well with the scientific achievements in this area.

The article includes:

- extensive and correct literature review (up-to-date),

- correct measures and methods of multidimensional data analysis, applied in the correct order (Cronbach's alphas, Ward's method, squared Euclidean distance, and ANOVA),

- discussion with other studies in the area (importantly, some studies are confirmed, some are extended, and some are in polemics with the presented studies).

The only thing I could suggest is to extend the area where this research is relevant. YAC caring for parents or grandparents is the implementation of intergenerational transfers. Maybe it is worth adding such a keyword.

Author Response

# Reviewer 3

I highly appreciate the text submitted for review.

Care for a sick, elderly or disabled person is often the responsibility of the family members of that person. Notably, state actions are not able to release society from this activity. I even think it would be inappropriate. That is why it is so important to analyze the situation of carers (mental health also). This research fits very well with the scientific achievements in this area.

The article includes:

- extensive and correct literature review (up-to-date),

- correct measures and methods of multidimensional data analysis, applied in the correct order (Cronbach's alphas, Ward's method, squared Euclidean distance, and ANOVA),

- discussion with other studies in the area (importantly, some studies are confirmed, some are extended, and some are in polemics with the presented studies).

The only thing I could suggest is to extend the area where this research is relevant. YAC caring for parents or grandparents is the implementation of intergenerational transfers. Maybe it is worth adding such a keyword.

Thank you for your comments. As recommended, we mentioned the intergenerational transfers as an object of future research by mentioning the finding of Hamill (2012) which explains that the more a parent cares for a grandparent, the more the young carers will care for her/him too (cf. p.14, lines 552-558).

Reviewer 4 Report

The subject of the paper is important, however following issues should be addressed. Therefore, I recommend major revision.

1) The paper needs to be proof-readed for one more round in terms of spelling rules and punctuation.

2) The research question was not clear. It should be emphasized in the introduction section by clear statements.

3) Why is there a need for investigating health level of YACs? It should be clearly explained in the introduction section.

4) Literature gap and how this paper could fill the gap is not clear because of lack of literature review section. It should be addressed.

5) Policy proposals, made to whom, and how it could be done was not clear. Authors should pay more attention to policy recommendations, and how it could be done should be identified with a clear statement.

Author Response

# Reviewer 4

The subject of the paper is important, however following issues should be addressed. Therefore, I recommend major revision.

1) The paper needs to be proofread for one more round in terms of spelling rules and punctuation.

As recommended, the paper has been proofread one more time by an English expert.

2) The research question was not clear. It should be emphasized in the introduction section by clear statements. 

The present study section has been revised in accordance with the comment of the Reviewer 2 (cf. p.3, lines 107-127). The research question has been revised as follows: “This study applies a person-oriented approach to examine the diversity of caring experiences through caring activities and their links to mental health for emerging adult students.”

3) Why is there a need for investigating health level of YACs? It should be clearly explained in the introduction section.

As suggested, we added a specific paragraph in the introduction section to highlight the need for investigating mental health, as YACs are at risk of experiencing poor mental health (cf. p.2, lines 89-92 and p.3, lines 93-94).

4) Literature gap and how this paper could fill the gap is not clear because of lack of literature review section. It should be addressed. 

We hope that the modification in line with your comments and those of Reviewers 1 and 2 provides a clearer picture of how this paper could fill a literature gap. In addition, we insisted on the importance of considering emerging adult students’ caring experiences in a person-oriented approach, in order to better understand their experience as a whole and, consequently, to better support them (cf. p.3, lines 107-124).

5) Policy proposals, made to whom, and how it could be done was not clear. Authors should pay more attention to policy recommendations, and how it could be done should be identified with a clear statement.

Thank you for your comment. As recommended, we added an Implications section in our manuscript (cf. p.15, lines 579-606).

Round 2

Reviewer 2 Report

I highly appreciate the revised manuscript, well written and referenced, for review. 

Reviewer 4 Report

The authors have addressed great part of queries. Therefore, editorial can consider the paper for potential publication.